# Nucleobase synthesis in interstellar ices

Yasuhiro Oba [1]*, Yoshinori Takano [2,5], Hiroshi Naraoka [3,4], Naoki Watanabe [1] & Akira Kouchi [1]

The synthesis of nucleobases in natural environments, especially in interstellar molecular clouds, is the focus of a long-standing debate regarding prebiotic chemical evolution. Here we report the simultaneous detection of all three pyrimidine (cytosine, uracil and thymine) and three purine nucleobases (adenine, xanthine and hypoxanthine) in interstellar ice analogues composed of simple molecules including $H_2O$, CO, $NH_3$ and $CH_3OH$ after exposure to ultraviolet photons followed by thermal processes, that is, in conditions that simulate the chemical processes accompanying star formation from molecular clouds. Photolysis of primitive gas molecules at 10 K might be one of the key steps in the production of nucleobases. The present results strongly suggest that the evolution from molecular clouds to stars and planets provides a suitable environment for nucleobase synthesis in space.

[1] Institute of Low Temperature Science (ILTS), Hokkaido University, N19W8, Kita-ku, Sapporo, Hokkaido 060-0819, Japan. [2] Department of Biogeochemistry, Japan Agency for Marine-Earth Science and Technology (JAMSTEC), 2-15 Natsushima, Yokosuka, Kanagawa 237-0061, Japan. [3] Department of Earth and Planetary Sciences, Kyushu University, 744 Motooka, Nishi-ku, Fukuoka, Fukuoka 819-0395, Japan. [4] Research Center for Planetary Trace Organic Compounds, Kyushu University, 744 Motooka, Nishi-ku, Fukuoka, Fukuoka 819-0395, Japan. [5] Present address: Biogeochemistry Program, Japan Agency for Marine-Earth Science and Technology (JAMSTEC), 2-15 Natsushima, Yokosuka, Kanagawa 237-0061, Japan. *email: oba@lowtem.hokudai.ac.jp

Nucleobases play an essential role in the biology of terrestrial organisms since they are the basic units used to record genetic information. From the viewpoint of origins of life on the Earth, therefore, they have been the target for laboratory experiments on the prebiotic synthesis in terrestrial environments[1–3]. The formation of nucleobases can also take place even in extraterrestrial environments, as evidenced by the detection of these species in carbonaceous meteorites[4,5]. Given that the exogenous delivery of organic molecules to Earth by meteorites and comets during the late heavy bombardment period before 3.8 billion years ago might have played a role in constraining the breadth of the initial inventory of organic molecules present on the early Earth[6], understanding the formation process of extraterrestrial molecules is of special importance for deciphering the chemical evolution before the birth of life on Earth. A number of laboratory studies have been performed to gain a better understanding of the molecular evolution that takes place during the process of formation of stars ($T > 100\,K$) from molecular clouds ($T \sim 10\,K$). As a result of these studies, it is known that many abundant molecules in interstellar ices such as $H_2O$, $CH_3OH$ and $NH_3$ can be effectively produced by non-energetic surface reactions that occur in the typical conditions found in molecular clouds[7]. Moreover, larger, more complex organic molecules such as amino acids and sugars can be produced from a mixture of those simpler molecules after exposure to ultraviolet (UV) photons and cosmic ray analogues (e.g. protons and electrons) followed by heating to room temperature[8–15]. This evidence suggests that these processes might be involved in the prebiotic synthesis of complex organic molecules in space[15,16]. By contrast, although several studies reported the prebiotic synthesis of nucleobases from formamide ($NH_2CHO$)[17,18], ammonium cyanide ($NH_4CN$)[2] and urea ($CO(NH_2)_2$)[19,20] under relatively warm conditions (i.e. near or above room temperature), there are no reports on the formation of nucleobases from abundant molecules in interstellar ices through the combination of photolysis at astrophysically relevant low temperatures and subsequent thermal processes. In particular, due to the limited number of related studies[14], the mechanism of formation of nitrogen-containing heterocyclic rings including purine and pyrimidine, which are the basic structures of nucleobases and have been utilised in previous studies for nucleobase production[21–24], from a mixture of abundant molecules in interstellar ices remains controversial. To understand the prebiotic evolution of biologically important chemical species in molecular clouds, the development of a synthetic pathway mimicking that process would be of pivotal importance.

With this aim, here we show experimental results on the formation of nucleobases under astrophysically relevant conditions, whereby a mixture of simple molecules ($H_2O:CO:NH_3:CH_3OH = 5:2:2:2$) is exposed to UV photons on a reaction substrate at 10 K. Subsequently, the products obtained are analysed using a high-resolution mass spectrometer (HRMS) coupled with a high-performance liquid chromatograph (HPLC). The most notable result of our experiments is the detection of all DNA/RNA nucleobases except guanine in the same sample.

## Results: Identification of nucleobases in the reaction product

Various kinds of nitrogen heterocycles containing more than two nitrogen atoms including six nucleobases (cytosine, uracil, thymine, adenine, hypoxanthine and xanthine) were identified in the organic residues isolated following the experiments described in the Methods section. The structures of these molecules are reported in Fig. 1. Figure 2 shows the mass chromatograms of a uracil standard and that of a recovered organic sample with the mass-to-charge ratio ($m/z$) of 113.0346, corresponding to

protonated uracil, which was analysed by an HPLC equipped with a reversed-phase C18 separation column (see Methods). The presence of uracil in the recovered sample was confirmed by the peak indicated by solid arrows in Fig. 2b, c. The identification of other five nucleobases (cytosine, thymine, adenine, hypoxanthine and xanthine) and uracil under different analytical condition in the sample is evidenced in Fig. 3 and Supplementary Figs. 1−6. No peaks attributable to any of the nucleobases were observed in the blank sample (Supplementary Fig. 7). By comparison with the peak area of the external standards, whose concentrations were adjusted from several tens of $ng\,\mu l^{-1}$ to $1\,\mu g\,\mu l^{-1}$, the carbon-based yield of nucleobases relative to the deposited gas mixtures was determined to be 1–4 parts per million (ppm) for pyrimidine nucleobases and 38–136 parts per billion (ppb) for purine nucleobases (Table 1). In the recovered organic sample, we also detected other molecules of prebiotic interest such as nitrogen heterocycles, amino acids and dipeptides, whose corresponding data are included in the Supplementary information (Supplementary Figs. 8–20, Supplementary Table 1). The presence of nucleobases and other molecules was reconfirmed by additional analyses performed under different analytical conditions, in which another separation column was used on the HPLC/HRMS system (Table 1) (see Methods for detailed information). To the best of our knowledge, there has been no report on the detection of both pyrimidine and purine nucleobases from the same astrophysical ice analogues after exposure to UV photons at 10 K. In previous studies, both types of nucleobases were not identified in the same sample where pyrimidine (Fig. 1, compound (9)) or purine (Fig. 1, compound (11)) was used as the initial reactant[21–24]. In addition, we found that the use of isotopically labelled molecules such as $^{15}NH_3$ and deuterated methanol isotopologues (e.g. $CH_2DOH$ and $CH_3OD$) led to the formation of $^{15}N$- and D-substituted nucleobases, respectively (see Fig. 4 and Supplementary Figs. 21–29). These results confirm that the nucleobases detected are not contaminants but produced from the experimental reactants.

## Discussion

Note that we did not perform experiments under conditions specifically aimed at the production of nucleobases, such as those using pyrimidine or purine as an initial reactant. Instead, we employed similar photo- and thermochemical conditions to those used for over three decades to simulate interstellar ice conditions[8–10,12,13]. Therefore, our results indicate that nucleobases would have been produced also in past experiments; however, they were not detected in those pioneering studies most likely due to the lack of appropriate analytical methods for the unambiguous identification of those molecules in complex mixtures. Recent advances in HRMS and chromatography with high sensitivity have provided the means to identify with extremely high precision a specific molecule at the isotopologue level in a complex mixture of organic molecules[25–27]. The detection of nucleobases under two different analytical conditions further confirms their presence in organic residues produced from a mixture of simple molecules.

Another possible key factor for the detection of nucleobases may be the temperature at which the photolysis takes place. Thus, nucleobases could not be detected in photochemically and thermally processed ice samples, despite using similar experimental conditions to those employed in the present study with the exception of several parameters[27]. The most striking difference between both experimental conditions is the temperature at which the deposited samples were exposed to UV photons (77 K in their study versus 10 K in the present study). Although previous studies suggest that the temperature of photolysis might not have a significant effect on the composition of the organic residues[10], the

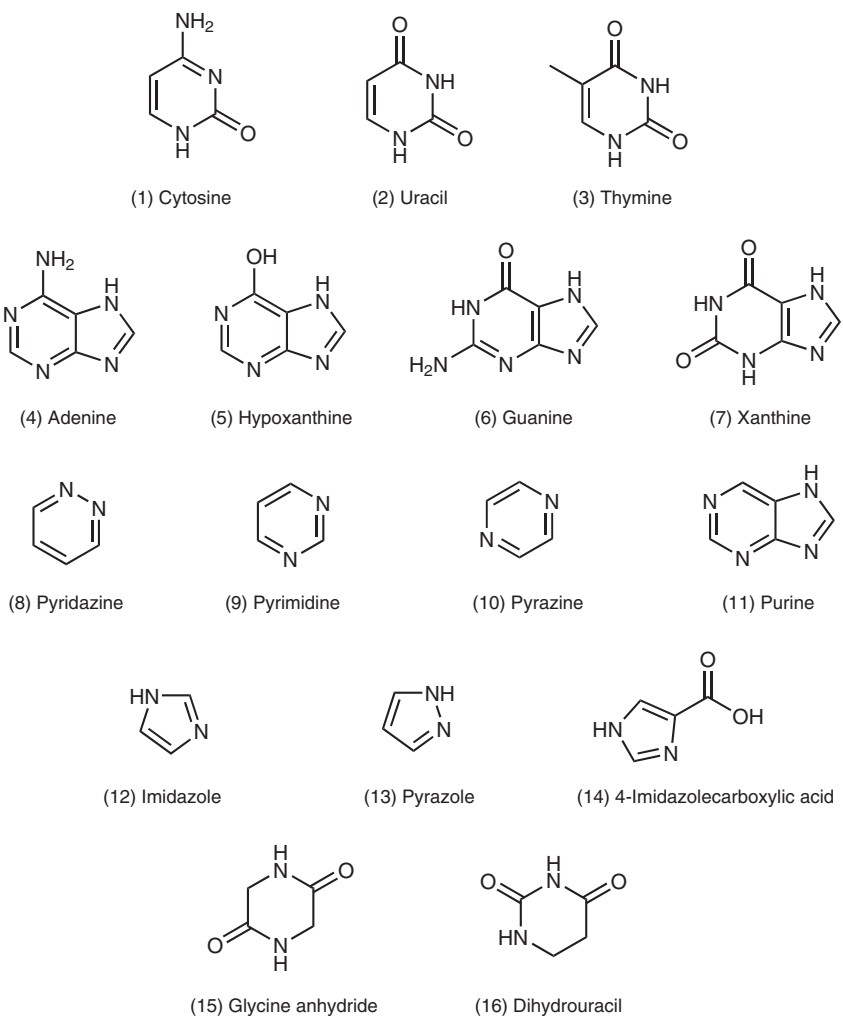

**Fig. 1** Names and structures of the products detected (except guanine) in the organic residues. The numbers in parentheses are used to identify the relevant compounds in Table 1, where the yield of each species is reported

chemistry that takes place in ice should be affected by temperature[26]. Since CO cannot adsorb on the reaction substrate at 77 K, this compound was not used in previous experiments performed at such temperature[10,12,27]. A presumable role of CO in the formation of nucleobases or their precursors might explain the failure to detect any nucleobases in the previous study[27] (see Supplementary Note 2 for further discussion). In addition, UV-exposed interstellar ice analogues are assumed to possess lower viscosity than amorphous solid water at temperatures above 65 K[28], a feature that could potentially cause different chemical reactivity patterns at different temperatures. Sugars and their related molecules were abundant (more than several hundreds of ppm)[12] in organic residues that were produced via photolysis of interstellar ice analogues ($H_2O$: $CH_3OH$:$NH_3 = 10$:3.5:1) at 78 K, and were analysed using two-dimensional gas chromatography time-of-flight mass spectrometry. However, these molecules were not positively identified in the sample prepared at 10 K (Supplementary Figs. 30–33), and only the upper limit is provided (e.g. <12 ppm for ribose, Supplementary Table 2). Apart from the differences in the experimental and analytical procedures, the striking differences on sugar formation compared with the previous report suggest that the photolysis temperature may also play a role in the formation of sugars and nucleobases in organic residues.

Several theoretical studies have been performed to elucidate the prebiotic formation of nucleobases[29–32]. For example, it was proposed that the prebiotic formation of both pyrimidine and purine nucleobases (except thymine) initiates from the reaction of hydrogen cyanide (HCN) with water[30]. Also, thymine formation was proposed to begin with the reaction of isocyanic acid (HNCO) with propanal ($CH_3CH_2CHO$)[32], both of which have been detected in the interstellar medium[33,34] and in organic residues produced in similar processes[10,35]. However, there has been no consensus on the synthetic pathways towards each nucleobase under interstellar conditions, which is mainly due to the lack of experimental evidence that supports the formation of nucleobases via these pathways. In the experiments conducted in previous studies on the formation of nucleobases starting from pyrimidine, pyrimidine nucleobases were not a major product when pyrimidine and other molecules such as $H_2O$, $CH_4$ or $CH_3OH$ were exposed to UV photons at 14–30 K[21,22]. Since the inefficient formation of nucleobases observed in previous studies was interpreted as resulting from the difficult introduction of substituents on the pyrimidine substrate via radical addition reactions, the detection of only a trace amount of synthesised pyrimidine in the present study suggests that the proposed pyrimidine-based formation route would not be dominant. With regard to the formation of purine nucleobases, their abundance was two orders of magnitude lower than that of purine in the same sample, e.g. adenine/purine $= 5 \times 10^{-2}$ (Table 1). This ratio is also orders of magnitude higher than the purine nucleobase

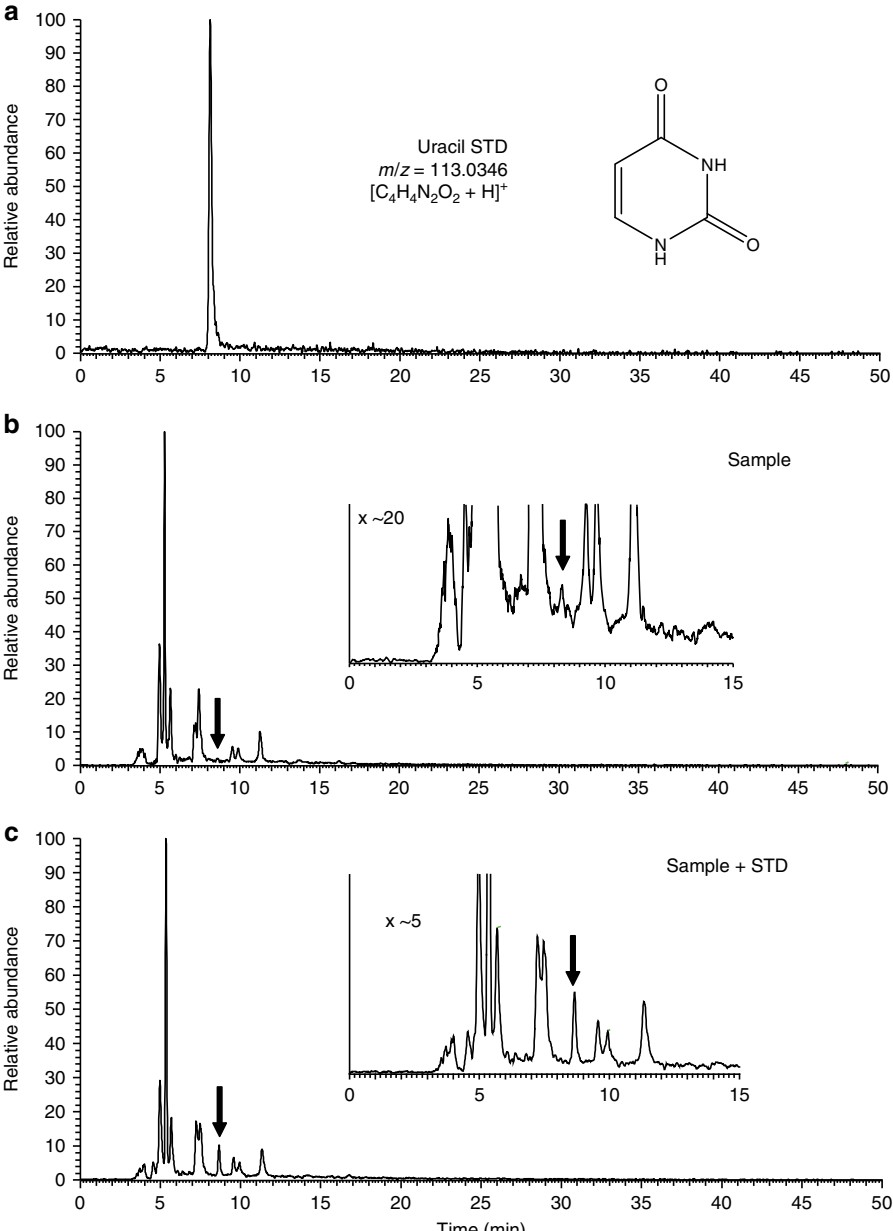

**Fig. 2** Identification of uracil in the organic residues. Mass chromatograms of **a** the uracil standard, **b** the analyte sample and **c** the co-injected mixture of uracil standard and analyte sample at a mass-to-charge ratio ($m/z$) of 113.0346. A C18 separation column was used for the analysis by HPLC/HRMS. The solid arrow indicates the presence of uracil. The inset shows an enlarged spectrum from 0 to 15 min. The intensity of the peak at ~8.6 min, which is associated with uracil (panel **a**), is higher in the co-injected sample (panel **c**) than in the analyte-only sample (panel **b**)

yield from a purine substrate in previous experiments (e.g. hypoxanthine/purine $< 3 \times 10^{-3}$)[24]. Hence, similarly to the case of pyrimidine nucleobases, it seems reasonable to rule out a purine-based pathway for the formation of purine nucleobases from purine under the present experimental conditions. Further experimental and computational studies are highly required to decipher the formation pathways of nucleobases under interstellar conditions.

Assuming that the present experimental results provide a good indication of the inventory of nucleobases formed in the interstellar medium, one can conclude that there is a good relation between interstellar and meteoritic nucleobases in terms of molecular distributions (e.g. the detection of both pyrimidine and purine nucleobases in the synthesised organic residues and in carbonaceous meteorites). In addition, the yield of nucleobases (~10 ppm in total, Table 1) relative to that of amino acids (~1000 ppm in total, Supplementary Table 1), which is ~$10^{-2}$ in the present study, is roughly in agreement with the relative abundance of those molecules in meteorites, which was found to be, for example, 180 ppb for nucleobases[4] and 11 ppm for amino acids[36] in the Murchison meteorite. Since various processes could be involved in the modification of the chemical compositions that occurs in the evolution from the interstellar environments to the solar system formation, the observed agreement in the relative abundances may not necessarily constrain the origin of meteoritic nucleobases. Nevertheless, nucleobases, if they are formed in the interstellar medium could be partly preserved at environments that have not experienced extensive alteration processes, and eventually be incorporated as an organic inventory in planetary systems. Hence, the first detection of nucleobases and a variety of their structural isomers in

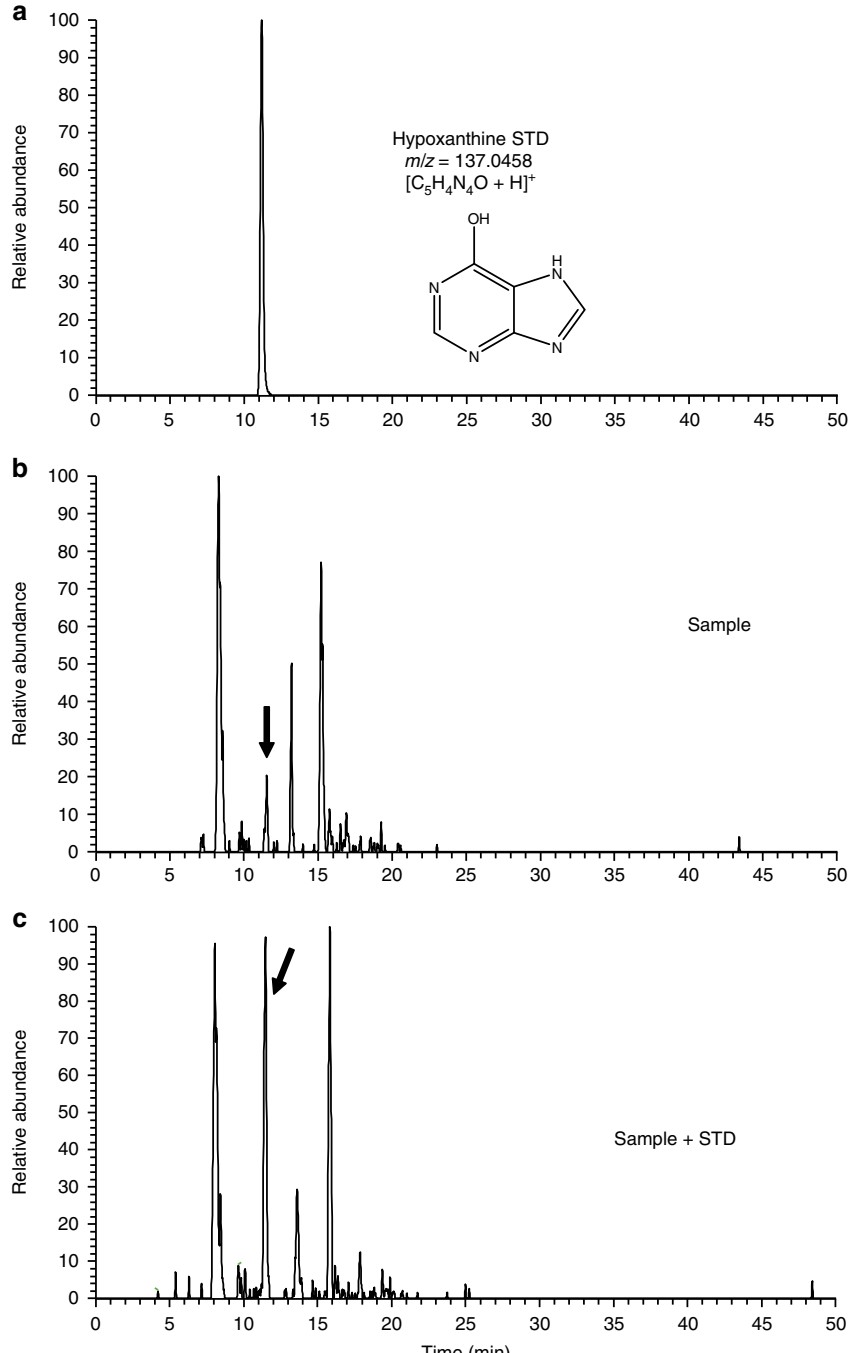

**Fig. 3** Identification of hypoxanthine in the organic residues. Mass chromatograms of **a** the hypoxanthine standard, **b** the analyte sample and **c** the co-injected mixture of hypoxanthine standard and analyte sample at a mass-to-charge ratio (*m/z*) of 137.0458. A C18 separation column was used for the analysis by HPLC/HRMS. The solid arrow indicates the presence of hypoxanthine. The intensity of the peak at ~11.3 min, which is associated with hypoxanthine (panel **a**), is higher in the co-injected sample (panel **c**) than in the analyte-only sample (panel **b**)

processed interstellar ice analogues has important consequences for the definition of the chemical inventory present during the early stage of chemical evolution in space. These findings may open new avenues for future related studies such as astronomical observations, chemical modelling and laboratory experiments.

## Methods

**Apparatus and UV irradiation experiments**. The photolysis of interstellar ice analogues was performed using a setup for analysis of molecular and radical reactions of astrochemical interest (SAMRAI)[25,26]. SAMRAI includes an ultra-high vacuum reaction chamber, a Fourier transform infrared spectrometer (FTIR), a quadrupole mass spectrometer and an aluminium (Al) substrate that can be cooled to 10 K using a helium (He) refrigerator. The base pressure is of the order of $10^{-7}$ Pa. Two deuterium ($D_2$) discharge lamps (L12098, Hamamatsu Photonics), whose photon fluxes are in the range $10^{13}$–$10^{14}$ photons $cm^{-2} s^{-1}$, are attached to the reaction chamber. Photon fluxes are estimated using the electric current measured by a photodiode (AXUV199G, Opto Diode Corp.) placed in front of the substrate. Gaseous samples of $H_2O$, CO, $NH_3$, and $CH_3OH$ in a mixing ratio of 5:2:2:2, respectively, were supplied via continuous vapour deposition (gas deposition rate of $\sim 5 \times 10^{12}$ molecules $cm^{-2} s^{-1}$) with photoirradiation onto the Al substrate for ~200 h at 10 K. The 200-h-long deposition yields the ice with the thickness of ~3600 monolayers (ML; 1 ML = $1 \times 10^{15}$ molecules $cm^{-2}$) if no photolysis of molecules occurred under the conditions utilised in this study. The photon fluence for 200 h corresponds to $\sim 2 \times 10^{7-8}$ years in molecular clouds when the photon

**Table 1 Quantification results for the nitrogen heterocyclic molecules targeted in the present study**

| Name of molecule | Molecular formula | Molecular structure | Mass-to-charge ratio ($m/z$) of the protonated ion | Yield (ppm)[a] by a C18 column | Yield (ppm)[a] by a Hypercarb™ column |
|---|---|---|---|---|---|
| **Nucleobases** | | | | | |
| Cytosine | $C_4H_5N_3O$ | (1) | 112.0505 | 2 | 1 |
| Uracil | $C_4H_4N_2O_2$ | (2) | 113.0346 | 1 | 4 |
| Thymine | $C_5H_6N_2O_2$ | (3) | 127.0502 | 2 | <3 |
| Adenine | $C_5H_5N_5$ | (4) | 136.0618 | 0.1 | — |
| Hypoxanthine | $C_5H_4N_4O$ | (5) | 137.0458 | 0.06 | 0.2 |
| Guanine | $C_5H_5N_5O$ | (6) | 152.0567 | — | — |
| Xanthine | $C_5H_4N_4O_2$ | (7) | 153.0407 | 0.04 | — |
| **Nitrogen heterocycles** | | | | | |
| Pyridazine | $C_4H_4N_2$ | (8) | 81.0447 | 19 | 31 |
| Pyrimidine | $C_4H_4N_2$ | (9) | 81.0447 | <1 | <1 |
| Pyrazine | $C_4H_4N_2$ | (10) | 81.0447 | 35 | 41 |
| Purine | $C_5H_4N_4$ | (11) | 121.0509 | 2 | 5 |
| Imidazole | $C_3H_4N_2$ | (12) | 69.0447 | 1152 | 1163 |
| Pyrazole | $C_3H_4N_2$ | (13) | 69.0447 | 89 | 20 |
| 4-Imidazolcarboxylic acid | $C_4H_4N_2O_2$ | (14) | 113.0346 | b | 139 |
| Glycine anhydride | $C_4H_6N_2O_2$ | (15) | 115.0502 | 3 | 42 |
| Dihydrouracil | $C_4H_6N_2O_2$ | (16) | 115.0502 | <1 | 61 |

For backing up the quantitative evaluation of those N-containing target molecules, we conducted the two independent chromatographic separations and co-injection determination with the corresponding authentic standard reagent (Supplementary Note 1). The small scale detection and calibration lines of the orbitrap mass spectrometry were also validated as shown in Supplementary Fig. 35
[a]Relative weight with relevance to the total deposited gas in part per million (1 ppm = 0.0001%) normalised with each carbon abundance
[b]Positively identified but not quantified

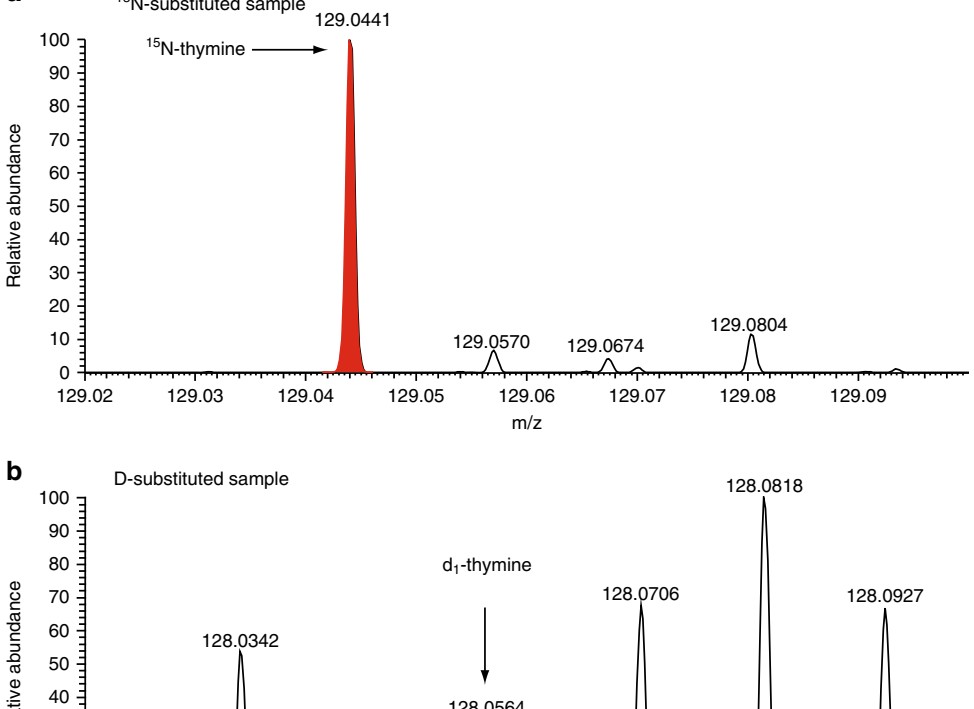

**Fig. 4** Identification of thymine in the isotopically labelled organic residues. Mass spectra of thymine observed in the **a** ¹⁵N- and **b** D-substituted samples at $m/z$ from 129.02 to 129.1 and from 128.02 to 128.10, respectively. Red coloured peaks at the $m/z$ of 129.0441 in panel (**a**) and at the $m/z$ of 128.0564 in panel (**b**) correspond to the ¹⁵N-substituted (¹⁵N–) thymine ($C_5H_6{}^{15}N_2O_2$: the $m/z$ of the protonated ion is 129.0443) and the singly deuterated ($d_1$–) thymine ($C_5H_5DN_2O_2$: the $m/z$ of the protonated ion is 128.0565), respectively. Please see the Methods section and the Supplementary Figs. 21–29 for further details on the ¹⁵N- and D-isotope probing experiments

flux is $10^4$ photons $cm^{-2} s^{-1}$, which is a mean photon flux in those environments[37]. The number of photons per molecule is in the range of 2–20. In a separate experiment, a mixture of deuterated methanol isotopologues including $CH_3OH$, $CH_2DOH$, $CHD_2OH$, $CD_3OH$, and $CH_3OD$ was used instead of pure $CH_3OH$ in a mixing ratio of 100:30:6:1:2, respectively, which is similar to the ratio observed in the low-mass protostar IRAS16293-2422 (ref. [38]). $CH_3OD$ was introduced via a separate gas line to avoid hydrogen isotopic exchange of the –OD group with other polar molecules (i.e., $H_2O$ and $NH_3$) before vapour deposition. This experiment would be helpful to estimate the deuterium enrichment of nucleobases which could be present in that environment. In addition, we performed an additional $^{15}N$-isotope probing experiment under the same conditions with the use of $^{15}NH_3$ gas ($^{15}N$ purity = 98%) instead of $^{14}NH_3$. A similar procedure was repeated without using gaseous samples to check a potential contribution from inside the chamber. The recovered sample was analysed as an entire procedural blank in this experiment (Supplementary Fig. 7).

**Molecular identification of the experimental products**. After simultaneous gas deposition and photon irradiation, the substrate was warmed up to room temperature to remove volatile species at a ramping rate of $0.2 K min^{-1}$, and the formation of solid organic residues was confirmed using FTIR[19]. The organic residues were dissolved in several tens of microlitres of a water/methanol mixture (1/1 by vol./vol.) and extracted from the reaction substrate using a small amount of quartz wool. The quartz wool with the samples was further transferred into a separate glass vial, and 0.5 ml of $H_2O$ was added to the vial. Subsequently, the aqueous solution was collected using a microsyringe and transferred into another glass vial. For an accurate chromatographic baseline resolution when focusing on the target molecules, especially those with nitrogen-containing functional groups (i.e., amide, amino, imino, and N-heterocyclic species; please see the Supplementary Fig. 34), a purification procedure was performed using cation-exchange column chromatography (AG-50W-X8 resin; 200–400 mesh, Bio-Rad Laboratories)[39]. This pretreatment advantageously eliminates uncharacterised matrix effects to assist with the evaluation of the irradiation products. The purified solution was dried under a gentle nitrogen gas ($N_2$) flow and subsequently dissolved in 50 µl of ultrapure $H_2O$ (QToFMS grade from Wako Chemical, Ltd.) before analysis. All glassware and the quartz wool were heated in air at 450 °C for 3 h to prevent contamination by organic compounds.

The sample solution was injected into an orbitrap mass spectrometer (Q Exactive Plus, Thermo Fischer Scientific) with a mass resolution of $m/\Delta m = \sim 140{,}000$ at a mass-to-charge ratio ($m/z$) of 200 via a high-performance liquid chromatograph (HPLC) system (UltiMate 3000, Thermo Fischer Scientific) equipped with a reversed-phase C18 separation column ($1.5 \times 250$ mm, particle size of 3 µm, InertSustain C18, GL Science) at 40 °C. The eluent program for this HPLC setup is as follows: solvent A ($H_2O$ + 0.1% formic acid by volume), solvent B (acetonitrile + 0.1% formic acid by volume) = 100:0 for the initial 5 min, followed by a linear gradient of A:B = 40:60 at 35 min, and it was kept at this ratio for 10 min. The flow rate was $70 µl min^{-1}$.

Nucleobases were also analysed using the same HPLC/HRMS equipped with a Hypercarb™ separation column ($4.6 \times 150$ mm, particle size of 5 µm, Thermo Fischer Scientific) at 10 °C to verify their presence in the sample. The eluent program is as follows: at $t = 0$, solvent A (20 mM nonafluoropentanoic acid in distilled water + 0.1% formic acid (dissolved)), solvent B (acetonitrile + 0.1% formic acid (dissolved)) = 100:0, followed by a linear gradient of A:B = 40:60 at $t = 60$ min and it was kept at this ratio for 10 min. The flow rate was $0.2 ml min^{-1}$. The detailed analytical conditions of the HPLC system have been described previously[25,40].

The identification of nitrogen-bearing molecules was based on a co-injection analysis in which the analyte sample and standard reagent solutions were analysed as part of the same HPLC/HRMS run.

For sugar molecules, we analysed the organic residues without the cation-exchange chromatography purification procedure (cf. this section for amide, amino, imino, and N-heterocyclic species). The mass spectra were recorded in the positive electrospray ionisation (ESI) mode for the nucleobases, other nitrogen heterocycles, dipeptides, and amino acids with an $m/z$ range of 50–400 and a spray voltage of 3.5 kV. To analyse the sugars, the mass spectra were recorded in the negative ESI mode with an $m/z$ range of 50–215 and a spray voltage of 3.0 kV. The capillary temperature of the ion transfer was 300 °C. The injected samples were vaporised at 300 °C. We set up an inverse gradient program to maintain the ionisation efficiency during the ESI. To minimise analytical noise and the background signals in the liquid chromatography (LC) and orbitrap mass spectrometry (MS), we used high purity grade water and acetonitrile (LC/MS grade from Wako Chemical, Ltd.). Under these experimental conditions, the mass precision is always better than 3 ppm for each chromatogram (e.g. $113.0348 \pm 0.0003$ for protonated uracil). The same volume of distilled water was measured to validate the contamination level of the MS; no prebiotic molecules were detected during the analytical blank measurement. For another sample (processed as described earlier in this paragraph), when no gases were deposited (i.e., UV photons only) on the substrate, we found that no prebiotic molecules formed (Supplementary Fig. 7), further confirming the blank control. Standard reagent grade target molecules were used and amino acids were purchased from Wako Chemicals, Ltd. and Sigma-Aldrich, Ltd. Pyrimidine and purine nucleobases and other nitrogen heterocycles were purchased from Tokyo Chemical Industry, Ltd.

and Sigma-Aldrich, Ltd., and dipeptides were from AnaSpec, Inc. and PH Japan, Ltd. Dihydrouracil was synthesised in our laboratory from reagent grade β-alanine and potassium cyanate (both from Wako Chemicals, Ltd.), according to the method reported by Dakin[41]. The target molecules (standards) were dissolved in distilled $H_2O$, and a mixture of the solution was analysed using the aforementioned procedure. To prevent misidentification, structural isomers were not analysed during the same run; e.g., to identify pyrimidine and its isomers (pyridazine and pyrazine), we performed three independent analyses.

After the analyses of the standard reagents and the sample in a separate run when the C18 column was used, 1 µl (5 µl when the Hypercarb™ was used) of the sample solution was co-injected with 1 µl (2 µl for the Hypercarb™) of a mixture of the standard reagents. The concentration of the co-injected solution was adjusted to not significantly exceed the sample peak heights. Eddhif et al.[27] identified various prebiotic molecules based on a comparison of the retention times of their target molecules on a separation column. Although this method is appropriate when the number of peaks in the mass chromatogram is not high, it becomes unsuitable when a number of peaks appear very close to each other. Moreover, the co-injection analysis is not always able to identify molecules in a mass chromatogram due to an insufficient separation of peaks and/or the absence of an appropriate chemical reagent. In this case, analysis of the sample using a different separation column on the HPLC/HRMS would be ideal to firmly identify a target molecule in a mixture of complex organic molecules. In the present study, we identified pyrimidine nucleobases by using two different separation columns (C18 and Hypercarb™) and the yields that were estimated under the two different conditions were consistent with each other, which strongly supports our conclusion that pyrimidine nucleobases are actually present in the sample. In contrast, under the present analytical conditions, purine nucleobases, except for hypoxanthine, were not eluted from the Hypercarb™ separation column probably due to the strong interaction between nucleobases and the stationary phase of the column; their detection was successful when the C18 column was used as a separation column. Hence, the cross-validation analysis is a promising approach to better understand the molecular compositions in complex organic residues.

## Data availability

The data that support the findings of this study are available from the corresponding author (Y.O.) upon reasonable request.

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

## Acknowledgements

We thank Dr. M. Hashiguchi (Kyushu University) for her technical supports on the sample analysis by the orbitrap MS. We also thank Drs. N. Ohkouchi and S. Furota (JAMSTEC) for the discussion of nitrogen molecules. Drs. T. Hama and W.M.C. Sameera (ILTS, Hokkaido University) are acknowledged for the discussion on the formation of nucleobases in interstellar environments. This work was partly supported by MEXT KAKENHI Grant Number JP25108006 and JSPS KAKENHI Grant Numbers JP15H05749, JP16H04083, and JP17H04862.

## Author contributions

Y.O. designed this study in consultation with N.W. and A.K. Y.O. performed photochemical reactions to synthesise organic residues including nucleobases and other molecules. Y.T. extracted and purified the target molecules from the organic residues. Y.O., Y.T. and H.N. performed the analysis of organic molecules using an orbitrap MS coupled with an HPLC. Y.O. wrote the manuscript. All the authors commented on the paper.

## Competing interests

The authors declare no competing interests.

## Additional information



