## [Peer Review File · Nature Communications]

Reviewers' comments:

Reviewer #1 (Remarks to the Author):

The authors make very exciting and bold claims about the formation of all biological nucleobases (except guanine) from simple ice mixtures. The results are novel and could be of significant interest to the astrobiology community. Based on the current state of the work, however, the claims are not yet fully justified and additional experiments should be conducted before publication.

1. A major issue with this manuscript is that many of the positive results were not verified by isotopic studies and therefore contamination cannot be ruled out. The authors attempted to perform isotopic labeling experiments, but did not confirm the presence of labeled nucleobases with the possible exception of Thymine and some other non-nucleobases. Experiments should be repeated to demonstrate that all of the nucleobases were formed in the experiment.

Additionally, the choice of isotopic studies is somewhat inexplicable. If the authors wanted to use deuterium as a tracer, the reactants (including H₂O and NH₃) should have been fully deuterated to avoid confusion rather than an arbitrary mixture of CH₃OH, CH₂DOH, CHD₂OH, CD₃OH, and CD₃OD. If it was the authors intention to study deuterium fractionation specifically as is reported in the supplementary material, that should have been a separate experiment that also used deuterated H₂O and NH₃ in realistic relative abundances (the ice mixture for this experiment is not particularly realistic).

Notably, Figure S21 is used as a demonstration that deuterated thymine is formed. This figure is insufficient. The y-axis does not appear to be meaningful so it is difficult to visually compare each thymine variant. More importantly, to prove that this is deuterated thymine, an actual mass spectrum should be shown rather than chromatograms. This mass spectrum should be contrasted with the mass spectrum of ordinary thymine and it should be demonstrated that the peaks associated with deuterated thymine are separate and resolvable from nearby peaks caused by natural ¹³C in ordinary thymine.

Finally, an alternative tracer such as ¹⁵NH₃ would have made a superior tracer for nucleobases because it is not solvent exchangeable, further ensuring the strength of the results.

2. The ice mixture chosen in this manuscript is by no means unusual and has been used in previous experimental attempts to produce nucleobases with no positive results. The authors claim that the

source of the breakthrough of this study is the use of HPLC/HRMS for the analysis, but a previous study conducting similar experiments with the same analysis technique did not yield nucleobases (Eddhif et al. 2018). The authors conclude that CO, which could not be condensed in the Eddhif study might play a key role in the formation of nucleobases, but make no attempt to verify this (e.g. use ^{13}CO as a tracer).

3. The following statement is inaccurate: “how nitrogen-containing heterocyclic rings including purine and pyrimidine (i.e., the basic structure of nucleobases), which were utilised in previous studies for the aim of nucleobase production, are produced from a mixture of the simple molecules remains altogether unclear.”

While far from settled science, there are hypotheses for how nitrogen heterocycles may have been produced (e.g., production in stellar outflows through C_2H_2 and HCN polymerization).

Minor issues:

1. A paper was just published in this journal that discusses the formation of sugars and should be cited: Nuevo, M., Cooper, G. and Sandford, S. A. (2018) Deoxyribose and deoxysugar derivatives from photoprocessed astrophysical ice analogues and comparison to meteorites. *Nature Communications*, volume 9, Article number: 5276

2. The English in the manuscript should be cleaned up. There are numerous examples of extraneous words and non-standard usage. Two examples from page 1:

“The formation of nucleobases can take place even in the extraterrestrial environments...”

“... to gain a better understanding of the molecular evolution that takes place during the process whereby stars are formed (Temp. >100 Kelvin) from molecular clouds ($T \sim 10\text{K}$).”

Reviewer #2 (Remarks to the Author):

The authors have reported nucleobases and other nitrogenated heterocycles from UV exposure of relevant astrophysical ice analogs for the first time. This is clearly an important accomplishment and will undoubtedly be interesting to the broader community. However, more detail is needed to evaluate the molecular yields and conclusions of the experimental results. The analytical instrumentation is state-of-the-art, but the validation and statistical analysis is not yet described adequately in the manuscript.

Major comments:

Methods: In order to make strong conclusions about the yields of the detected species (or upper limits of detected species), it is important to have done the validation studies for each molecule. Did you validate your methods, including the purification procedure with each of the columns, for each of the molecular species? How was the quantification conducted? Did you plot calibration curves to determine linearity and calculate the limit of quantification for each of these species? What are the errors in each reported yield measurement?

In particular, did all samples undergo purification with the cation-exchange material? It is possible that the low yields of sugars (glyceraldehyde, glycoaldehyde dimer, etc.) are simply due to their low retention in the purification step. If this is the case, then the absence of sugars may be an experimental artifact, and further implies that the peak identifications in S27 and S28 may be incorrect and the conclusion in lines 110-115 may not be supported.

Minor comments:

Lines 40-44: "By contrast, there are no reports on the formation of nucleobases from a mixture of the simple molecules through a combination of photochemical and thermal processes. In particular, how nitrogen-containing heterocyclic rings including purine and pyrimidine ... are produced from a mixture of the simple molecules remains altogether unclear."

The phrase "a mixture of the simple molecules" used in both of these sentences is somewhat ambiguous, and should be clarified. It is true that no work so far has demonstrated pyridines and pyrimidines from UV exposure of the precise mixture of H₂O:CO:NH₃:CH₃OH ices or from relevant mixtures of astrophysically-relevant ice. But several other purines and/or pyrimidines have been reported from thermal and photochemical exposure of simple mixtures [UV irradiation of urea/H₂O/C₂H₂ [Shanker, Bhushan et al. 2011] and in heated formamide/H₂O solutions [Barks, Buckley et al. 2010; Menor-Salván and Marín-Yaseli 2013]]. They have also been synthesized in the absence of UV from hydrogen cyanide and ammonia at temperatures from 27-100 °C [Oró and Kimball 1961], and from ammonium cyanide [Oró 1960; Levy, Miller et al. 1999], in the gas phase [Hamid, Bera et al. 2014], and from other thermal or photochemical reactions that are also arguably "simple" [Hayatsu, Studier et al. 1972; Voet and Schwartz 1982].

Lines 74-76 “This is the first report to detect the both pyrimidine and purine nucleobases in the same sample, which was not possible in previous studies where pyrimidine or purine was used as the initial reactant.”

See previous comment; please clarify that it is the first report to detect both pyrimidine and purine nucleobases in the same UV-irradiated astrophysical ice sample (or something to this effect).

Lines 110-115 Reference 12 used a different gas mixture and a different deposition pressure and flux – please specify. The differences in results may be due to temperature, but the other conditions should still be noted if different from yours.

Line 111 “photolysis of gas mixtures” implies that the chemistry is only happening in the gas phase – perhaps say “UV irradiation during deposition” or something similar instead.

Table 1 – * and *** symbols are used but ** symbol indicating “Not identified mainly due to analytical problems” does not appear in the table. Also, please state what analytical problems were encountered.

Line 258 – how was the thickness determined?

Line 261 “The photon fluence for 200 h corresponds to $\sim 2 \times 10^6$ years in molecular clouds when the photon flux is 10^5 photons $\text{cm}^{-2}\text{s}^{-1}$ in those environments. The photon to gas flux ratio is therefore in the range of 2-20.” Please remove “therefore” since the calculation of the ratio doesn’t follow directly from the previous sentence. I would also specify “photon to deposited molecule ratio” or “number of photons per molecule” instead of the more ambiguous “photon to gas flux ratio”.

Please check that pyridine is spelled properly throughout (not “pyradine”). Similarly, there glycerinaldehyde is spelled incorrectly in Fig. S25 and glucose is misspelled in the caption of S28.

Line 310-311: Does this also mean that your extraction windows are <3 ppm for each of your chromatograms? Please state explicitly somewhere.

Consider merging Table S1 with Table 1 or at least moving into the main text. For the glycerinaldehyde, glycoaldehyde dimer, ribose, and glucose please also explain whether these were also attempted with the Hypercarb column and consider adding the “upper limit” estimates to Table 1/S1.

Barks, H. L., R. Buckley, G. A. Grieves, E. Di Mauro, N. V. Hud and T. M. Orlando (2010). "Guanine, Adenine, and Hypoxanthine Production in UV-Irradiated Formamide Solutions: Relaxation of the Requirements for Prebiotic Purine Nucleobase Formation." *ChemBioChem* 11(9): 1240-1243.

Hamid, A. M., P. P. Bera, T. J. Lee, S. G. Aziz, A. O. Alyoubi and M. S. El-Shall (2014). "Evidence for the Formation of Pyrimidine Cations from the Sequential Reactions of Hydrogen Cyanide with the Acetylene Radical Cation." *The Journal of Physical Chemistry Letters* 5(19): 3392-3398.

Hayatsu, R., M. H. Studier, S. Matsuoka and E. Anders (1972). "Origin of organic matter in early solar system—VI. Catalytic synthesis of nitriles, nitrogen bases and porphyrin-like pigments." *Geochimica et Cosmochimica Acta* 36(5): 555-571.

Levy, M., S. L. Miller and J. Oró (1999). "Production of guanine from NH₄ CN polymerizations." *Journal of molecular evolution* 49(2): 165-168.

Menor-Salván, C. and M. R. Marín-Yaseli (2013). "A New Route for the Prebiotic Synthesis of Nucleobases and Hydantoins in Water/Ice Solutions Involving the Photochemistry of Acetylene." *Chemistry – A European Journal* 19(20): 6488-6497.

Oró, J. (1960). "Synthesis of adenine from ammonium cyanide." *Biochemical and biophysical research communications* 2(6): 407-412.

Oró, J. and A. P. Kimball (1961). "Synthesis of purines under possible primitive earth conditions. I. Adenine from hydrogen cyanide." *Archives of Biochemistry and Biophysics* 94(2): 217-227.

Shanker, U., B. Bhushan, G. Bhattacharjee and Kamaluddin (2011). "Formation of Nucleobases from Formamide in the Presence of Iron Oxides: Implication in Chemical Evolution and Origin of Life." *Astrobiology* 11(3): 225-233.

Voet, A. B. and A. W. Schwartz (1982). "Uracil synthesis via HCN oligomerization." *Origins of life* 12(1): 45-49.

Reviewer #3 (Remarks to the Author):

This paper reports the synthesis of nucleobases in analogues of interstellar ices containing H₂O:CO:CH₃OH:NH₃ exposed to ultraviolet radiation. This is a new finding and has important implications for prebiotic chemistry in space, I only know of a work by Meierhenrich et al. (2004), *Chem. Eur. J.* that reports the formation of a few N-heterocycles in a similar experiment, this work was not cited by the authors in the submitted manuscript. The paper is well written and easy to read.

Unfortunately, there is a good number of identified species that are in my opinion not sufficiently supported by the data, for which the signal-to-noise ratios are not reported. In my opinion, these would be at most tentative identifications with upper limits.

Already in Fig. 2, the potential uracil peak is very small, would benefit from estimation of the signal-to-noise ratio and I would regard as a tentative identification and give upper limits. This peak becomes clear and has the expected peak shape only after co-injection of the uracil standard. The same holds in my opinion for Figs. S1, S3A (I don't even see a peak in the inlet of panel (b) Sample), S3B, S4A, S4B (also no peak, here there is no amplification), S9A (I see no peak in panel b), S9B (in panel b the narrow points to a region that cannot be regarded as a detection), S13A (would benefit from an amplification), S15A (very minor peak?), S16A, S18A, S18B, (panel b, just noise?),

In Table 1, there is a huge gap in the estimated abundances of imidazole, pyrazole, pyrazine, pyridazine, and the other products. In my opinion, it would have been wise to separate the assignments in three groups, one for the identified species, another one with tentative identifications with upper limits, and finally one with non-detections when no peak was apparently observed.

I can therefore not accept this paper for publication.

Small remarks:

Table 1. "Pyradine" should be pyrazine.

Page 17: Give a reference for photon flux in molecular clouds, they report 10^5 photons $\text{cm}^{-2} \text{s}^{-1}$. Shen et al. 2004 give a value of 10^4 .

Replies to comments by Reviewers

We appreciate the constructive reviews from three reviewers on our manuscript (NCOMMS-19-03496-T) entitled “Nucleobase synthesis in interstellar ices”. We carefully read the whole comments and modified the original version of the manuscript based on their helpful comments. The changes we made based on the reviewer’s comments are noted in **red colour font** in the revised manuscript/supporting information. Our replies to each comment are denoted following to the reviewer’s comments.

Reviewer #1

The authors make very exciting and bold claims about the formation of all biological nucleobases (except guanine) from simple ice mixtures. The results are novel and could be of significant interest to the astrobiology community. Based on the current state of the work, however, the claims are not yet fully justified and additional experiments should be conducted before publication.

[Reply] Thank you very much for your positive comments on our manuscript. Point-by-point based replies to his/her comments are shown below. We have performed additional ^{15}N -isotope probing experiments (^{15}N purity = 98%) using $^{15}\text{NH}_3$ as suggested by the reviewer #1. In addition, we would like to present additional data on the identification of deuterated species in the original sample and recovery data of aliphatic, aromatic, and N-heterocyclic molecules through the column chromatography, which is definitely backing up our conclusion that the observed nucleobases are not contaminant but produced in the present experiment. I hope our replies can fulfill the reviewer’s requests.

Comment 1. A major issue with this manuscript is that many of the positive results were not verified by isotopic studies and therefore contamination cannot be ruled out. The authors attempted to perform isotopic labeling experiments, but did not confirm the presence of labeled nucleobases with the possible exception of Thymine and some other non-nucleobases. Experiments should be repeated to demonstrate that all of the nucleobases were formed in the experiment.

Additionally, the choice of isotopic studies is somewhat inexplicable. If the authors wanted to use deuterium as a tracer, the reactants (including H_2O and NH_3) should have been fully deuterated to avoid confusion rather than an arbitrary mixture of

CH₃OH, CH₂DOH, CHD₂OH, CD₃OH, and CD₃OD. If it was the authors intention to study deuterium fractionation specifically as is reported in the supplementary material, that should have been a separate experiment that also used deuterated H₂O and NH₃ in realistic relative abundances (the ice mixture for this experiment is not particularly realistic).

Notably, Figure S21 is used as a demonstration that deuterated thymine is formed. This figure is insufficient. The y-axes do not appear to be meaningful so it is difficult to visually compare each thymine variant. More importantly, to prove that this is deuterated thymine, an actual mass spectrum should be shown rather than chromatograms. This mass spectrum should be contrasted with the mass spectrum of ordinary thymine and it should be demonstrated that the peaks associated with deuterated thymine are separate and resolvable from nearby peaks caused by natural ¹³C in ordinary thymine.

Finally, an alternative tracer such as ¹⁵NH₃ would have made a superior tracer for nucleobases because it is not solvent exchangeable, further ensuring the strength of the results.

[Reply] We have clearly confirmed that nucleobases did not form in the blank experiment, as already shown in Fig. S7. The reason why we used a mixture of deuterated methanol isotopologues is not only to confirm that the products are not a contaminant, but also to estimate a possible deuterium fractionation upon the formation of nucleobases which may form in the actual interstellar environment, where molecules are deuterated in part (e.g. IRAS16293-1422, Parise et al. 2006). We added the reason to use the mixture of deuterated methanol isotopologues in the method section. To further strengthen our conclusion that the detected nucleobases are the real products in the present experiment, we would like to show the mass spectra of isotopically-labelled nucleobases in the D- or ¹⁵N-substituted products as the reviewer proposed.

Firstly, Fig. S21(a) in the revised SI shows the mass spectra of the peak appeared at ~8 min for the uracil standard reagent at the *m/z* of 113.0 to 114.1. The C18 separation column was used for the analysis. The peak at 113.0346 corresponds to the protonated ion of non-deuterated (d₀) uracil. The insets show enlarged spectra at the *m/z* of 114.026 to 114.046, which corresponds to the *m/z* range for singly deuterated (d₁) uracil. Very small peaks attributed to the ¹³C-substituted uracil (denoted as ¹³C-uracil) and d₁-uracil were observed in the inset at the *m/z* of 114.0380 and 114.0407, respectively. On the other hand, the abundance of the d₁-uracil isotopolog was clearly enhanced in the mass spectra for the

D-substituted sample. It should be noted that the peak height for the d₁-uracil in the D-substituted sample (Fig. 21b) significantly exceeds that for the ¹³C-uracil, which is unlikely to occur in natural environments. Hence, this is the clear evidence to show that the identified uracil is the product in the present experiment where deuterated methanol was used as a reactant. The same is also true for other pyrimidine nucleobases cytosine and thymine (Figs. S22 and S23, respectively). As for purine nucleobases, unfortunately, we were unable to detect their deuterated counterparts in the D-substituted sample probably due to their low abundances. We then would like to replace the figures for the mass chromatograms of thymine (Fig. S21), imidazole (Fig. S22), pyridazine (Fig. S23), and 4-imidazolecarboxylic acid (Fig. S24) with the mass spectra of uracil (as Fig. S21), cytosine (as Fig. S22), and thymine (as Fig. S23) detected in the D-substituted sample. Mass spectra for other deuterated species were not shown in the revised version. Accordingly, a degree of deuterium fractionation originally reported in Table S2 and the description about the deuterium fractionation were deleted from the Supporting Information. Those results will be published in our future publications.

Secondly, we performed additional experiments on the formation of ¹⁵N-substituted nucleobases under the same experimental conditions except the use of ¹⁵NH₃ instead of ¹⁴NH₃, as the reviewer proposed. As shown in Figs. S24-S29 in the revised version, we observed the mass peaks derived from nucleobase isotopologs whose nitrogen atoms were all replaced with ¹⁵N in the mass spectra of the ¹⁵N-substituted sample. Note that due to the lack of the standard reagents for the ¹⁵N-substituted nucleobases, we did not perform the co-injection analysis and hence did not quantify them. The identification of the target molecules was performed by the comparison of the retention times of the peaks and isotopic mass shift of ¹⁵N between the sample and the standard reagents. Therefore, these isotopically labelled ¹⁵N-experimental results clearly exclude a possibility that the observed nucleobases are due to contaminants.

Hence, based on the above results, we are confident that the identified molecules are the products in the present experiment. We added the mass spectra of D-substituted and ¹⁵N-substituted thymine in each isotopically labelled sample as Fig. 4 in the main text.

Comment 2. The ice mixture chosen in this manuscript is by no means unusual and has been used in previous experimental attempts to produce nucleobases with no positive results. The authors claim that the source of the breakthrough of this study is the use of HPLC/HRMS for the analysis, but a previous study conducting similar experiments with the same analysis technique did not yield nucleobases (Eddhif et al. 2018). The authors conclude that CO, which could not be condensed in the Eddhif study might play a key

role in the formation of nucleobases, but make no attempt to verify this (e.g. use ^{13}C as a tracer).

[Reply] We do not aim at elucidating the formation pathways of nucleobases in the present study; rather, what we would like to emphasize is that nucleobases can be produced under the typical icy conditions in molecular clouds. The role of CO is one of the possible explanations based on the comparison of the experimental conditions and results between Eddhif et al. (2018) and the present study. Note that we do not exclude other possibilities to cause the observed difference between the two studies; e.g. differences in UV photon flux/fluence, gas compositions except CO, gas deposition rate, etc. In particular, we expect that the difference in photolysis temperature (77 K vs. 10 K) may have a strong contribution to the observed differences between the two studies. As already mentioned in the main text, at >65 K, the radicals formed by photolysis might be able to diffuse more efficiently in the ice than that at 10 K, which could have resulted in the suppression of nucleobase formation, somehow. While, the formation of other molecules such as sugars and larger amino acids (e.g. glutamic acid and histidine), which were below the detection limit in our samples, could have been enhanced at such higher temperature ($\sim 77\text{K}$). In the present paper, we would like to show decisive evidence that nucleobases can be produced from simple gas molecules under the astrophysically relevant, low-temperature icy conditions. Detailed studies on the formation pathways of nucleobases as well as the possible effect of photolysis temperatures and the role of CO are beyond the scope of the present study but will be reported in our future outcomes. Then, we added some words to imply there could be other possibilities except the photolysis temperature to cause the observed difference between the two studies.

Comment 3. The following statement is inaccurate: “how nitrogen-containing heterocyclic rings including purine and pyrimidine (i.e., the basic structure of nucleobases), which were utilised in previous studies for the aim of nucleobase production, are produced from a mixture of the simple molecules remains altogether unclear.”

[Reply] We modified the last part of the sentence as follows: “In particular, due to the limited number of the related studies, the mechanism of formation of nitrogen-containing heterocyclic rings including purine and pyrimidine, which are the basic structures of nucleobases and have been utilised in previous studies for nucleobase production²¹⁻²⁴, from a mixture of simple molecules remains controversial.”

Minor comment 1. 1. A paper was just published in this journal that discusses the formation of sugars and should be cited: Nuevo, M., Cooper, G. and Sandford, S. A. (2018) Deoxyribose and deoxysugar derivatives from photoprocessed astrophysical ice analogues and comparison to meteorites. Nature Communications, volume 9, Article number: 5276

[Reply] Added as suggested in the lines of 36-39 and the reference list (ref. 13).

Minor comment 2. The English in the manuscript should be cleaned up. There are numerous examples of extraneous words and non-standard usage. Two examples from page 1:

“The formation of nucleobases can take place even in the extraterrestrial environments...”

“... to gain a better understanding of the molecular evolution that takes place during the process whereby stars are formed (Temp. >100 Kelvin) from molecular clouds (T~10K).”

[Reply] Since the English in the manuscript was edited by native English speakers, we thought that it does not matter on the quality of English. However, we accept the reviewer’s opinion and then the manuscript has been again carefully reviewed by an experienced editor whose first language is English and who specializes in editing papers written by scientists whose native language is not English (see a certificate of editing in the next page). Since there a number of modifications associated with the English grammar, etc., they are not highlighted in the text. We have carefully checked that the English modifications did not change the scientific contents in the present paper.

CERTIFICATE OF EDITING

This is to certify that the paper titled Nucleobase synthesis in Interstellar Ices commissioned to us by Yasuhiro Oba (北海道大学) has been edited for English language, grammar, punctuation, and spelling by Enago, the editing brand of Crimson Interactive Pvt. Ltd under Normal Editing.

ISO/IEC 27001:2013 Certified

ISO 9001:2015 Certified

Issued by: Enago, Crimson Interactive Pvt. Ltd.
1001, Techniplex - II, S. V. Road,
Goregaon (W), Mumbai 400062, India.
Phone: 03-5050-5374
Fax: 03-4496-4934

Disclaimer: The author is free to accept or reject our changes in the document after our editing. However, we do not bear responsibility for revisions made to the document after our edit on **28 May 2019**.

Global www.enago.com, www.ulatus.com, www.voxtab.com
Japan www.enago.jp, www.ulatus.jp, www.voxtab.jp
Taiwan www.enago.tw, www.ulatus.tw
China www.enago.cn, www.ulatus.cn
Brazil www.enago.com.br, www.ulatus.com.br

Germany www.enago.de
Russia www.enago.ru
Arabic www.enago.ae
Turkey www.enago.com.tr
S. Korea www.enago.co.kr

About Crimson:
Crimson Interactive Inc. provides
English language editing, transcription,
and translation services to individuals
and corporate customers worldwide.

Reviewer #2

The authors have reported nucleobases and other nitrogenated heterocycles from UV exposure of relevant astrophysical ice analogs for the first time. This is clearly an important accomplishment and will undoubtedly be interesting to the broader community. However, more detail is needed to evaluate the molecular yields and conclusions of the experimental results. The analytical instrumentation is state-of-the-art, but the validation and statistical analysis is not yet described adequately in the manuscript.

[Reply] Thank you very much for the reviewer's positive comments on our manuscript. We would like to further present detailed evidence to support our conclusion, which we hope can fulfill the reviewer's requests.

Major comment. Methods: In order to make strong conclusions about the yields of the detected species (or upper limits of detected species), it is important to have done the validation studies for each molecule. Did you validate your methods, including the purification procedure with each of the columns, for each of the molecular species? How was the quantification conducted? Did you plot calibration curves to determine linearity and calculate the limit of quantification for each of these species? What are the errors in each reported yield measurement?

In particular, did all samples undergo purification with the cation-exchange material? It is possible that the low yields of sugars (glyceraldehyde, glycoaldehyde dimer, etc.) are simply due to their low retention in the purification step. If this is the case, then the absence of sugars may be an experimental artifact, and further implies that the peak identifications in S27 and S28 may be incorrect and the conclusion in lines 110-115 may not be supported.

[Reply] We have validated our experimental procedure for all target molecules in the present study. In particular, the standard reagents of the target molecules were treated in the same analytical procedure including the cation-exchange column chromatography (Fig. S34 with the unpublished data); we conducted the validation of the chromatographic recovery (%) by using various aliphatic, aromatic, N-heterocyclic molecules. As an individual verification after the ref. 41, it is important to note that the recovery of N-containing molecules showed better than $95.0 \pm 3.5\%$ (ave.) including the N-heterocyclic derivatives of -Proline, -Histidine, -Phenylalanine, -Tryptophan as model compounds (cf. Takano et al., 2019).

Fig. Examples of calibration curves for nucleobases measured by HPLC/HRMS. Left: Uracil, right: hypoxanthine.

We confirmed that our procedure is very suitable for extracting those molecules from a mixture of complex molecules; namely, the low yields of nucleobases and sugars are not due to the purification procedure. The quantification was performed by the comparison of the peak area between the sample and the standard reagent whose concentration is adjusted. We also had an analytical validation that the linearity between the injected amount and the peak area is guaranteed (see the attached graphs for the calibration curves of uracil and hypoxanthine as examples) within the range of the sample concentration (on the order of tens of pico gram per injection). The analytical error on the HPLC/HRMS is better than 10%. The limit of quantification for the standard reagent of nucleobases is in general on the order of pmol ($= 10^{-12}$ mol) or less under the present analytical conditions. However, the detection limit for nucleobases in our sample strongly depends on the degree of peak separation on the mass chromatogram with other structural isomers. In this regard, the yields for a couple of molecules were regarded as the upper limit in the revised version of the manuscript (see Tables 1 and S1 in the revised version).

Supplementary reference [41]: Y. Takano, S. Furota, N. O. Ogawa, and N. Ohkouchi: Analytical development of underivatized amino acids and peptide molecules. *Abstract for Annual Meetings of the Meteoritical Society* (2019).

Minor comment 1. Lines 40-44: "By contrast, there are no reports on the formation of nucleobases from a mixture of the simple molecules through a combination of photochemical and thermal processes. In particular, how nitrogen-containing heterocyclic rings including purine and pyrimidine ... are produced from a mixture of the

simple molecules remains altogether unclear.”

The phrase “a mixture of the simple molecules” used in both of these sentences is somewhat ambiguous, and should be clarified. It is true that no work so far has demonstrated pyridines and pyrimidines from UV exposure of the precise mixture of H₂O:CO:NH₃:CH₃OH ices or from relevant mixtures of astrophysically-relevant ice. But several other purines and/or pyrimidines have been reported from thermal and photochemical exposure of simple mixtures [UV irradiation of urea/H₂O/C₂H₂ [Shanker, Bhushan et al. 2011] and in heated formamide/H₂O solutions [Barks, Buckley et al. 2010; Menor-Salván and Marín-Yaseli 2013]]. They have also been synthesized in the absence of UV from hydrogen cyanide and ammonia at temperatures from 27-100 °C [Oró and Kimball 1961], and from ammonium cyanide [Oró 1960; Levy, Miller et al. 1999], in the gas phase [Hamid, Bera et al. 2014], and from other thermal or photochemical reactions that are also arguably “simple” [Hayatsu, Studier et al. 1972; Voet and Schwartz 1982].

[Reply] We appreciate the reviewer to let us know a number of the related studies performed previously. It is no doubt that these studies are very important for better understanding of the prebiotic evolution related to nucleobases. To our best knowledge, the present study is the first to show the formation of nucleobases by the photolysis of ices composed of “abundant molecules in interstellar ices” only at temperatures as low as “10 K”, as the reviewer mentioned. This is one of the key points that should be highlighted in the present study. Then, we refined some sentences in the introductory part of the manuscript to exemplify previous studies on the prebiotic formation of nucleobases and highlight the novelty of the achievement. Moreover, as for the use of “simple molecule”, we agree that it is actually ambiguous. Then, we replaced the words “simple molecules” in the line 41 with “abundant molecules in interstellar ices”. This expression should include H₂O, CO, NH₃, and CH₃OH but should not include C₂H₂, NH₂CHO, ammonium cyanide, HCN, purine, pyrimidine, etc, which were used in the previous studies. Accordingly, the words “Simple molecules” in the line 34 was replaced with “Many abundant molecules in interstellar ices”.

Minor comment 2. Lines 74-76 “This is the first report to detect the both pyrimidine and purine nucleobases in the same sample, which was not possible in previous studies where pyrimidine or purine was used as the initial reactant.”

See previous comment; please clarify that it is the first report to detect both pyrimidine and purine nucleobases in the same UV-irradiated astrophysical ice sample (or

something to this effect).

[Reply] The sentence was modified as follows: “To the best of our knowledge, this is the first report to detect both pyrimidine and purine nucleobases from the same astrophysical ice analogues after exposure to UV photons at 10 K. In previous studies, both types of nucleobases were not identified in the same sample where...”

Minor comment 3. Lines 110-115 Reference 12 used a different gas mixture and a different deposition pressure and flux – please specify. The differences in results may be due to temperature, but the other conditions should still be noted if different from yours.

[Reply] The gas composition and the analytical system applied in Meinert et al. (2016) were added into the text (see also the next reply). In addition, we modified the last sentence in the same paragraph as follows: “Apart from the differences in the experimental and analytical procedures, the striking differences on sugar formation compared with the previous report suggest that the photolysis temperature also plays a ...”.

Minor comment 4. Line 111 “photolysis of gas mixtures” implies that the chemistry is only happening in the gas phase – perhaps say “UV irradiation during deposition” or something similar instead.

[Reply] Since we do not think that the photolysis does not take place in the gas phase under their experimental conditions, the words “gas mixtures” were replaced with “interstellar ice analogues (H₂O, CH₃OH, and NH₃ = 10:3.5:1)”. In addition, the temperature (77 K) was modified appropriately (78 K).

Minor comment 5. Table 1 – * and *** symbols are used but ** symbol indicating “Not identified mainly due to analytical problems” does not appear in the table. Also, please state what analytical problems were encountered.

[Reply] We are very sorry that the use of “****” in Table 1 is not correct. This symbol was used in the earlier version of our manuscript but was not deleted in the final version. We then replaced the symbol “****” with “***”. Although not related to the present version of the manuscript, some other molecules were listed in the earlier version since their presence was confirmed by the co-injection method. However, due to the poor peak separation with

others, it was difficult to quantify those molecules. Hence, we explained such that.

Minor comment 6. Line 258 – how was the thickness determined?

[Reply] Under the present experimental conditions, the deposition rate of molecules was adjusted to $\sim 5 \times 10^{12}$ molecules $\text{cm}^{-1} \text{s}^{-1}$. This value was roughly estimated from the IR spectrum obtained in a separate test experiment. Then, since we deposited gases for ~ 200 h, the thickness was calculated as: $5 \times 10^{12} \times 60 \times 60 \times 200 = 3.6 \times 10^{18}$ molecules cm^{-2} . This value was divided by 1×10^{15} , which is equivalent to 1 monolayer (ML), and then we obtained 3600 ML.

Minor comment 7. Line 261 “The photon fluence for 200 h corresponds to $\sim 2 \times 10^6$ years in molecular clouds when the photon flux is 105 photons $\text{cm}^{-2} \text{s}^{-1}$ in those environments. The photon to gas flux ratio is therefore in the range of 2-20.” Please remove “therefore” since the calculation of the ratio doesn’t follow directly from the previous sentence. I would also specify “photon to deposited molecule ratio” or “number of photons per molecule” instead of the more ambiguous “photon to gas flux ratio”.

[Reply] The word “therefore” was deleted from here. In addition, we replaced the words “photon to gas flux ratio” with “number of photons per molecule”.

Minor comment 8. Please check that pyridine is spelled properly throughout (not “pyradine”). Similarly, there glycer-aldehyde is spelled incorrectly in Fig. S25 and glucose is misspelled in the caption of S28.

[Reply] The typos were corrected. We did not intend to show pyridine ($\text{C}_5\text{H}_5\text{N}$) but pyrazine ($\text{C}_4\text{H}_4\text{N}_2$). All “pyradine” were replaced with “pyrazine”.

Minor comment 9. Line 310-311: Does this also mean that your extraction windows are <3 ppm for each of your chromatograms? Please state explicitly somewhere.

[Reply] Yes, it is true. We modified the sentence as follows: “..., the mass precision is always better than 3 ppm for each chromatogram ...”

Minor comment 10. Consider merging Table S1 with Table 1 or at least moving into the main text. For the glycer-aldehyde, glycoaldehyde dimer, ribose, and glucose please

also explain whether these were also attempted with the Hypercarb column and consider adding the “upper limit” estimates to Table 1/S1.

[Reply] We have tried to analyze sugar and its related molecules by using the Hypercarb column under the similar conditions; however, they did not elute from the column because of very small scale experimental yields or optimization of ionization process on ESI was not sufficient for the sugar-related molecules. Please see the ref. 27 regards to the detection of sugar molecules (i.e., fructose, galactose, ribose, arabinose, xylose and isomers). We did not pursue further at the moment on the analysis of those molecules by using the Hypercarb column.

As for merging Table S1 with Table 1, the reviewer’s suggestion might be one of the options which we can select. However, we would prefer the present style since the present manuscript mainly focus on the formation of nucleobases and their related molecules (namely, nitrogen heterocycles). We do understand the quantification of other molecules is also important, but they might be outside the scope of the present study.

As for the upper limit for the yields of sugars, we will show those values in Table S2 (the original Table S2, which reported the degree of deuterium fractionation, was deleted from the revised version based on the other reviewer’s comment).

Reviewer #3

This paper reports the synthesis of nucleobases in analogues of interstellar ices containing H₂O:CO:CH₃OH:NH₃ exposed to ultraviolet radiation. This is a new finding and has important implications for prebiotic chemistry in space, I only know of a work by Meierhenrich et al. (2004), Chem. Eur. J. that reports the formation of a few N-heterocycles in a similar experiment, this work was not cited by the authors in the submitted manuscript. The paper is well written and easy to read.

[Reply] Thank you very much for the reviewer’s comments/criticism on our manuscript. We suppose the paper which the reviewer indicated may be Meierhenrich, Muñoz Caro, Schutte, Thiemann, Barbier, & Brack (2005), “Precursors of biological cofactors from ultraviolet irradiation of circumstellar/interstellar ice analogues”. Chem. Eur. J. 11, 4895-4900. This paper was included in the discussion for lines 36-39.

Comment 1. Unfortunately, there is a good number of identified species that are in my

opinion not sufficiently supported by the data, for which the signal-to-noise ratios are not reported. In my opinion, these would be at most tentative identifications with upper limits. Already in Fig. 2, the potential uracil peak is very small, would benefit from estimation of the signal-to-noise ratio and I would regard as a tentative identification and give upper limits. This peak becomes clear and has the expected peak shape only after co-injection of the uracil standard. The same holds in my opinion for Figs. S1, S3A (I don't even see a peak in the inlet of panel (b) Sample), S3B, S4A, S4B (also no peak, here there is no amplification), S9A (I see no peak in panel b), S9B (in panel b the narrow points to a region that cannot be regarded as a detection), S13A (would benefit from an amplification), S15A (very minor peak?), S16A, S18A, S18B, (panel b, just noise?).

[Reply] In many of Figures pointed out by the reviewer, we expanded further the inset to better show the presence of the target peak, as well as showing the degree of magnification in each inset. The signal-to-noise ratio is typically better than 9. Although the peak is small in most cases, based on the coinjection results, the most of peaks we assigned were surely derived from the target molecules. On the other hand, when we reanalyzed all mass chromatograms, we considered that some of the peak assignments might be insufficient due to their very low peak intensity. For example, we agree with the reviewer's comment that the detection of thymine in Fig. S4B (by Hypercarb) is not conclusive due to the low S/N (~ 2). Similarly, the detection of pyrimidine in Figs. S9A (by C18) and S9B (by Hypercarb), that of dihydrouracil in Fig. S16A (by C18), and that of glycylglycine in Fig. S18A (by C18) may be regarded as a tentative detection. Accordingly, for those analyses, each quantification result was modified as the upper limit (see Table 1).

Comment 2. In Table 1, there is a huge gap in the estimated abundances of imidazole, pyrazole, pyrazine, pyridazine, and the other products. In my opinion, it would have been wise to separate the assignments in three groups, one for the identified species, another one with tentative identifications with upper limits, and finally one with non-detections when no peak was apparently observed.

[Reply] This might be one of the options to show our quantification results. However, since we would like to focus on the detection of nucleobases in the present study, we would highlight those molecules, even if they are much less abundant compared to other molecules such as imidazole and pyridazine in terms of yields in the same sample.

Small remark 1. Table 1. "Pyradine" should be pyrazine.

[Reply] We replaced all “pyradine” with “pyrazine” in the manuscript.

Small remark 2. Give a reference for photon flux in molecular clouds, they report 10^5 photons $\text{cm}^{-2} \text{s}^{-1}$. Shen et al. 2004 give a value of 10^4 .

[Reply] If we use the value of 10^4 here, it corresponds to $\sim 2 \times 10^{7-8}$ years in those environments. If we use the value of 10^8 , which is considered to be equivalent to the interstellar UV flux, it leads to $\sim 2 \times 10^{3-4}$ years. The value of 10^5 used here was just an example for the photon flux to convert the photon fluence to time (or years); hence, we think that it is not appropriate to cite any references here.

Reviewers' comments:

Reviewer #1 (Remarks to the Author):

The authors have directly addressed most of my concerns about this manuscript and the addition of the ^{15}N work has made their results significantly more convincing. The paper is quite interesting and should be of value to the community. I only have one remaining quibble.

The language suggesting the photolysis temperature was “a key factor” in the detection of nucleobases is still too strong. The authors concede that they made no attempt to directly study the effects of conducting the photolysis at different temperatures and are making a hypothesis based on the comparison of their results to a previous study. The language should be adjusted to reflect that their assertion that temperature may play an important role is merely a hypothesis.

If the authors correct this, I have no further need to review the manuscript.

Reviewer #2 (Remarks to the Author):

The changes made by the authors have adequately addressed most of the points outlined, and all of the work that has been done to improve the manuscript is appreciated. My only remaining concern is the following:

Original comment: “In particular, did all samples undergo purification with the cation-exchange material? It is possible that the low yields of sugars (glyceraldehyde, glycoaldehyde dimer, etc.) are simply due to their low retention in the purification step...”

[Author Reply] “We have validated our experimental procedure for all target molecules in the present study. In particular, the standard reagents of the target molecules were treated in the same analytical procedure including the cation-exchange column chromatography (Fig. S34 with the unpublished data)”

Reviewer response: Thank you for the clarification. In the caption to Figure S34, please note which concentrations were used.

If the standards in all figures containing co-injected samples were subjected to the same analytical procedure, including purification step, please add a sentence to this effect in the Methods section.

If this was not done for all standards in the figures, then the recoveries of all relevant species at low concentrations under this procedure should be tested and added to Fig. 34. It is a good sign that several molecules showed better than 95% recovery, but one cannot assume that every other molecule would have similarly high recovery. Particularly, analytes of very low (ppb or ppm-level) concentrations can exhibit significantly lower recovery efficiencies, and if this was not accounted for already by subjecting the standards and their calibration solutions to identical treatment, the concentrations (and therefore some of the conclusions) may be skewed.

Reviewer #3 (Remarks to the Author):

Dear editor,

The manuscript has improved significantly after the corrections. I think this version is not confusing to the reader because the cases where a detection was not clear are regarded as upper limits. The quality of the work and the final results are certainly worth publication in this journal. It is of high relevance for a broad scientific community. I congratulate the authors for this work.

I just have a small clarification that the authors can take as optional: the UV field in the diffuse interstellar medium is indeed 10^8 photons $\text{cm}^{-2} \text{s}^{-1}$, but the more relevant value is 10^4 photons $\text{cm}^{-2} \text{s}^{-1}$, i.e. the secondary UV field in dense clouds where ice mantles are present, this value varies in circumstellar regions of YSOs. I have no personal interest in citing Shen et al. since I am not one of the authors, if you prefer you can cite the earlier reference of Cecchi-Pestellini & Aiello (1992) or leave the value of 10^5 photons $\text{cm}^{-2} \text{s}^{-1}$ without any mention to a given astrophysical environment.

Replies to comments by Reviewers

We greatly appreciate the constructive reviews from three reviewers on our manuscript (NCOMMS-19-03496-A) entitled “Nucleobase synthesis in interstellar ices”. We carefully read the whole comments and refined the 1st revised version of the manuscript based on their helpful comments. The changes we made based on the reviewer’s comments are noted in red colour font in the 2nd revised manuscript/supporting information. Our replies to each comment are denoted following to the reviewer’s comments.

Reviewer #1

[Reviewer’s comments]

The authors have directly addressed most of my concerns about this manuscript and the addition of the ¹⁵N work has made their results significantly more convincing. The paper is quite interesting and should be of value to the community. I only have one remaining quibble.

The language suggesting the photolysis temperature was “a key factor” in the detection of nucleobases is still too strong. The authors concede that they made no attempt to directly study the effects of conducting the photolysis at different temperatures and are making a hypothesis based on the comparison of their results to a previous study. The language should be adjusted to reflect that their assertion that temperature may play an important role is merely a hypothesis.

If the authors correct this, I have no further need to review the manuscript.

[Author reply]

We appreciate that the reviewer #1 is mostly satisfied with our replies to his/her previous comments. As for the point indicated above, we modified some sentences in the abstract and the discussion part (lines ~100-120) based on the reviewing comments.

Reviewer #2

[Reviewer's comments]

The changes made by the authors have adequately addressed most of the points outlined, and all of the work that has been done to improve the manuscript is appreciated. My only remaining concern is the following:

Original comment: "In particular, did all samples undergo purification with the cation-exchange material? It is possible that the low yields of sugars (glyceraldehyde, glycoaldehyde dimer, etc.) are simply due to their low retention in the purification step..."

[Author Reply] "We have validated our experimental procedure for all target molecules in the present study. In particular, the standard reagents of the target molecules were treated in the same analytical procedure including the cation-exchange column chromatography (Fig. S34 with the unpublished data)"

Reviewer response: Thank you for the clarification. In the caption to Figure S34, please note which concentrations were used. If the standards in all figures containing co-injected samples were subjected to the same analytical procedure, including purification step, please add a sentence to this effect in the Methods section. If this was not done for all standards in the figures, then the recoveries of all relevant species at low concentrations under this procedure should be tested and added to Fig. 34. It is a good sign that several molecules showed better than 95% recovery, but one cannot assume that every other molecule would have similarly high recovery. Particularly, analytes of very low (ppb or ppm-level) concentrations can exhibit significantly lower recovery efficiencies, and if this was not accounted for already by subjecting the standards and their calibration solutions to identical treatment, the concentrations (and therefore some of the conclusions) may be skewed.

[Author Reply] The authors appreciate the constructive comments. Firstly, the target molecules in the present study are N-containing molecules such as -amide, -amino, -imino, and N-heterocyclic species as we described in the initial manuscript. For further confirmation of the analytical procedure, we refined the context more precisely (lines 342-343) as follows:

"For comprehensive and quantitative evaluation between nitrogen containing molecules and sugar molecules, we analysed the organic residues without the isolation procedure by the cation-exchange chromatography for sugar-related compounds (cf. Method section for amide, amino, imino, and N-heterocyclic species). The mass spectra were recorded..."

Then, we would like to modify the previous replies to the reviewer #2 as “We have validated our experimental procedure for the target N-molecules in the present study”, which is consistent with our justification of initial declaration in the lines 315-319: “For an accurate chromatographic baseline resolution when focusing on the target molecules, especially those with nitrogen-containing functional groups (i.e., amide, amino, imino, and N-heterocyclic species; please see the supplementary information), a purification procedure was performed using cation-exchange column chromatography (AG-50W-X8 resin; 200–400 mesh, Bio-Rad Laboratories)³⁹.” The previous ref. 38 was currently modified to ref. 39 according to the comment from the reviewer #3.

Secondly, we note here that the validation ranges for the concentration and the corresponding recovery were in the scale of 200 ppm (average of recovery $>95.0 \pm 3.5$ %), 20 ppm ($>94.4 \pm 3.9$ %), 2 ppm ($>93.2 \pm 4.6$ %) through the cation-exchange chromatography with wet chemical controls (unpublished data: figure shown for the reviewing process, y-intercept $>90\%$ in 10 ppb scale, $R^2=0.96$) based on N-containing molecules.

Also, we added a sentence in the caption of Table S2 as follows:

“The determination of sugar-related compounds was conducted without the cation-exchange chromatography.”

Although we have been confident that the recoveries of all target molecules are fairly high ($>90\%$), we would like to refrain from showing those data on this paper because we would like to report them as an independent technical paper (i.e., ref. 46 with additional data for the improvement of refs. 39-40). If we report those data in this paper, a scientific impact on that future paper will be inevitably lowered.

Thirdly, we added the description of analytical process in the lines 195-197 with regards to the co-injected samples:

“For backing up the quantitative evaluation of those N-containing target molecules, we conducted the two independent chromatographic separations and co-injection determination with the corresponding authentic standard reagent (Supplementary Information). The small scale detection and calibration lines of the orbitrap mass spectrometry were also validated as shown in Figure S35.”

Reviewer #3

[Reviewer's comments]

The manuscript has improved significantly after the corrections. I think this version is not confusing to the reader because the cases where a detection was not clear are regarded as upper limits. The quality of the work and the final results are certainly worth publication in this journal. It is of high relevance for a broad scientific community. I congratulate the authors for this work.

I just have a small clarification that the authors can take as optional: the UV field in the diffuse interstellar medium is indeed 10^8 photons $\text{cm}^{-2} \text{s}^{-1}$, but the more relevant value is 10^4 photons $\text{cm}^{-2} \text{s}^{-1}$, i.e. the secondary UV field in dense clouds where ice mantles are present, this value varies in circumstellar regions of YSOs. I have no personal interest in citing Shen et al. since I am not one of the authors, if you prefer you can cite the earlier reference of Cecchi-Pestellini & Aiello (1992) or leave the value of 10^5 photons $\text{cm}^{-2} \text{s}^{-1}$ without any mention to a given astrophysical environment.

[Author reply]

Thank you very much for the reviewer's very positive comments and also for introducing an important paper by Cecchi-Pestellini & Aiello (1992). In the 2nd revised version of the manuscript, we cited the reference at the Method section as ref 37. Accordingly, the corresponding time scales in molecular clouds and the following reference numbers were modified appropriately.

[Ref. 37] Cecchi-Pestellini, C. & Aiello, S. Cosmic ray induced photons in dense interstellar clouds. *Mon. Not. R. Astron. Soc.* **258**, 125-133 (1992).

That's all for the replies.

REVIEWERS' COMMENTS:

Reviewer #2 (Remarks to the Author):

I appreciate the clarification in the text that the validation was not done for the sugar-related compounds, but find the new text to be somewhat confusing:

“For comprehensive and quantitative evaluation between nitrogen containing molecules and sugar molecules, we analysed the organic residues without the isolation procedure by the cation-exchange chromatography for sugar-related compounds (cf. Method section for amide, amino, imino, and N-heterocyclic species).”

- 1) If the validation was not done for the sugar-related compounds, you should not state that the sugar molecules have undergone a comprehensive and quantitative evaluation.
- 2) This is the first time mentioning “the isolation procedure,” instead perhaps call it just a cation-exchange chromatography purification procedure to match the terms used earlier in the text
- 3) The sentence is somewhat unclear overall. Perhaps reword the whole text to something like:

“For sugar molecules, we analyzed the organic residues without the cation-exchange chromatography purification procedure (cf. Method section for amide, amino, imino, and N-heterocyclic species).”

After this sentence is fixed, I have no further need to review the manuscript.

A final note of clarification that may be of future use to the authors regarding the recovery plot that was shown to maintain 90% recovery at very low concentrations:

Recovery efficiency can drop sharply and unexpectedly at low concentrations, and a perfectly linear extrapolation usually cannot be assumed. Often, these recoveries depend on the exact interactions of molecular groups, and therefore will also vary greatly from compound to compound (and therefore multiple plots may be needed).

Replies to comments by Reviewers

We greatly appreciate the constructive reviews from the reviewer #2 on our manuscript (NCOMMS-19-03496-B) entitled “Nucleobase synthesis in interstellar ices”. We carefully read the whole comments and refined the 2nd revised version of the manuscript based on their helpful comments. We modified a sentence as requested by the reviewer #2, as shown below. Our replies to each comment are denoted following to the reviewer’s comments.

Reviewer #2

[Reviewer’s comments]

I appreciate the clarification in the text that the validation was not done for the sugar-related compounds, but find the new text to be somewhat confusing:

“For comprehensive and quantitative evaluation between nitrogen containing molecules and sugar molecules, we analysed the organic residues without the isolation procedure by the cation-exchange chromatography for sugar-related compounds (cf. Method section for amide, amino, imino, and N-heterocyclic species).”

1) If the validation was not done for the sugar-related compounds, you should not state that the sugar molecules have undergone a comprehensive and quantitative evaluation.

2) This is the first time mentioning “the isolation procedure,” instead perhaps call it just a cation-exchange chromatography purification procedure to match the terms used earlier in the text

3) The sentence is somewhat unclear overall. Perhaps reword the whole text to something like:

“For sugar molecules, we analyzed the organic residues without the cation-exchange chromatography purification procedure (cf. Method section for amide, amino, imino, and N-heterocyclic species).”

After this sentence is fixed, I have no further need to review the manuscript.

[Author Reply] The authors appreciate the constructive comments. We modified the sentence as the reviewer proposed.

A final note of clarification that may be of future use to the authors regarding the recovery plot that was shown to maintain 90% recovery at very low concentrations:

Recovery efficiency can drop sharply and unexpectedly at low concentrations, and a perfectly linear extrapolation usually cannot be assumed. Often, these recoveries depend on the exact interactions of molecular groups, and therefore will also vary

greatly from compound to compound (and therefore multiple plots may be needed).

[Author Reply] Thank you very much for the comments. We understand the recovery can vary depending on the concentrations. However, at the concentration range applied in the present study, we have confirmed that the targeted nitrogen-bearing molecules can be measured quantitatively with high recovery.